# Ferritin Trajectories over Repeated Whole Blood Donations: Results from the FIND+ Study

**DOI:** 10.3390/jcm11133581

**Published:** 2022-06-21

**Authors:** Sara Moazzen, Maike G. Sweegers, Mart Janssen, Boris M. Hogema, Trynke Hoekstra, Katja Van den Hurk

**Affiliations:** 1Molecular Epidemiology Research Group, Max-Delbrück-Centrum für Molekulare Medizin in der Helmholtz-Gemeinschaft, 13125 Berlin, Germany; sara.moazzen@mdc-berlin.de; 2Donor Studies, Department of Donor Medicine Research, Sanquin Research, 1066CX Amsterdam, The Netherlands; m.sweegers@nki.nl; 3Transfusion Technology Assessment, Department of Donor Medicine Research, Sanquin Research, 1066CX Amsterdam, The Netherlands; m.janssen@sanquin.nl; 4Blood-Borne Infections, Department of Donor Medicine Research, Sanquin Research, 1066CX Amsterdam, The Netherlands; b.hogema@sanquin.nl; 5Department of Health Sciences or Amsterdam Public Health Research Institute, Vrije Universiteit Amsterdam, 1007MB Amsterdam, The Netherlands; trynke.hoekstra@vu.nl

**Keywords:** ferritins, blood, blood donors

## Abstract

Background: Depending on post-donation erythropoiesis, available iron stores, and iron absorption rates, optimal donation intervals may differ between donors. This project aims to define subpopulations of donors with different ferritin trajectories over repeated donations. Methods: Ferritin levels of 300 new whole blood donors were measured from stored (lookback) samples from each donation over two years in an observational cohort study. Latent classes of ferritin level trajectories were investigated separately using growth mixture models for male and female donors. General linear mixed models assessed associations of ferritin levels with subsequent iron deficiency and/or low hemoglobin. Results: Two groups of donors were identified using group-based trajectory modeling in both genders. Ferritin levels showed rather linear reductions among 42.9% of male donors and 87.7% of female donors. For the remaining groups of donors, steeper declines in ferritin levels were observed. Ferritin levels at baseline and the end of follow-up varied greatly between groups. Conclusions: Repeated ferritin measurements show depleting iron stores in all-new whole blood donors, the level at which mainly depends on baseline ferritin levels. Tailored, less intensive donation strategies might help to prevent low iron in donors, and could be supported with ferritin monitoring and/or iron supplementation.

## 1. Introduction

Repeated whole blood donations often result in low iron stores, potentially leading to donor deferral for low hemoglobin (Hb) and iron deficiency-related symptoms [1,2,3]. With the reporting of negative findings on the health and availability of donors, it is of paramount importance for blood collection centers to gain more insight into whole blood donors’ iron status and how low iron stores in blood donors should be managed.

Currently, international consensus on an appropriate policy for donor iron management is lacking [4]. Schotten et al. (2016) have found that ferritin levels on average require at least 180 days to return to pre-donation levels in a sample of 50 male whole blood donors aged 30–50 years [5]. On day 57, in only 32% of repeat donors and 25% of new donors ferritin levels returned to pre-donation levels. Optimal donation intervals may differ between donors, depending on the decrease in ferritin levels after a whole blood donation and an individual’s capability of restoring ferritin to pre-donation levels. This may be influenced by dietary and supplemental iron intake and other lifestyle behaviors, as well as by genetic traits that affect iron absorption and release [6].

Repeated measurements of ferritin levels over consecutive donations are generally lacking, as ferritin levels are only measured in cross-sectional studies or policies with limited measurements; at every 5th or 10th donation [7,8]. The prevalence of iron deficiency in repeat donors, even in those with normal Hb levels, is remarkably high [8,9,10,11,12]. In addition, once donors have become iron-deficient, their risk of subsequent Hb deferral is over tenfold increased [13]. Several investigations have assessed appropriate methods for lowering the rates of Hb deferrals, among which a recent study assessed changes in Hb levels through repeated donations to identify donors at risk for quick declines in Hb levels [14]. Previous studies have shown that Hb level recovery in donors depends on the availability of iron [15,16]; meanwhile, in the Netherlands, the general minimum intervals between whole blood donations for women and men are 122 and 56 days, respectively. Further sights into changes in ferritin levels over consecutive donations are needed to better understand how quickly iron deficiency develops, and whether or not this differs inter-individually; eventually, this could aid in the development of better-tailored donor deferral strategies. We, therefore, aimed at distinguishing ferritin trajectories based on measurements of ferritin levels from samples collected during consecutive whole blood donations over 2 years. Additionally, we investigated the risks of donors returning to donate with low iron levels, defined as either being iron-deficient or deferred for low Hb, concerning ferritin levels at the preceding donation.

## 2. Materials and Methods

### 2.1. Study Population

Sanquin is by law the only organization authorized for the collection and supply of blood (products) in The Netherlands. All donors undergo routine eligibility testing before donating, and plasma samples from every donation are stored for 2 years for quality control purposes. In January 2020, we retrospectively included 300 new donors, who donated at least twice in the two-year follow-up period from 11 September 2017 onwards, in the observational cohort study Ferritin measurement IN Donors—donor Population Longitudinal Study (FIND+). We only selected donors who donated at centers that did not implement Sanquin’s ferritin measurement policy before September 2019. Pseudonymized blood donor data, including age, sex, Hb levels, number of donations, height, and weight were obtained from the donor database (eProgesa software application; MAK-SYSTEMS, Paris, France). Samples were analyzed only from voluntary, non-remunerated, adult donors who provided written informed consent as part of routine donor selection and blood collection procedures. Sanquin’s Ethical Advisory Board reviewed and approved the study protocol.

### 2.2. Ferritin Measurements

Sanquin’s National Screening Laboratory routinely stores 850 µL of blood samples collected in K3EDTA tubes (Greiner Bio-One, Frickenhausen, Germany taken from the sampling pouch at each whole blood donation; these samples are pipetted into 96 well plates and stored for 2 years at −30 °C. One day before testing, plates containing selected samples were defrosted. The TECAN pipetting robot (TECAN Trading AG, Männedorf, Switzerland) was used for pipetting selected samples from wells to sample tubes, and ferritin levels were determined by Architect i2000sr (Abbott Diagnostics, Wiesbaden, Hessen Germany). For the validation of ferritin levels from stored samples, we compared ferritin levels from stored samples with ferritin levels measured in fresh samples (see Appendix A); this showed a slight underestimation of ferritin levels (−4.4 ng/mL on average) and differences between measurements mainly ranging from −30.1 to 21.3 ng/mL (limits of agreement). Measurement differences were larger with higher ferritin levels. At Sanquin, all donors undergo a screening visit before being invited to make their first donation. As ferritin levels were measured twice before the first donation, the mean ferritin level was used as a baseline level.

### 2.3. Statistical Analysis

The donor characteristics were described as medians and interquartile ranges, or as percentages in the case of proportions, stratified by gender. We applied growth mixture models to capture the longitudinal trajectories of ferritin levels and the variation in these trajectories among donors [17,18]. Growth mixture models are used to assign individuals to one of several subgroups (latent classes) based on temporal patterns in the data; in this case, donors with similar ferritin trajectories are assigned to one subgroup, and the subgroups are most different from each other in terms of ferritin trajectories.

We conducted the analyses in several steps. First, the heterogeneity of individual ferritin trajectories was visualized and assessed by plotting variations in ferritin levels through the donation time points for all study subjects (spaghetti plots). To identify clusters of donors with a similar pattern of ferritin levels over the repeated donation times within two years, we conducted growth mixture models [18,19,20,21]. We considered the ferritin levels as a dependent variable and donation numbers in two years (September 2017–September 2019) as the main independent variable. Given the fact that a finite set of polynomial functions of time could summarize the individual differences in trajectories, we fitted the first-order linear and second-order quadratic polynomial models. The Bayesian Information Criterion (BIC) was applied to compare the less complex model (i.e., a smaller number of latent classes) to the complex model (i.e., more classes) [22] and the logarithm of the Bayes factor (2*ΔBIC), where ΔBIC = BIC (complex) − BIC (less complex) [23]; moreover, when the absolute value of the logarithm of the Bayes factor 2*ΔBIC would indicate trivially (0–2), positive (2–6), strong (6–10), or very strong (>10), this provides the level of evidence for the null hypothesis that the less complex model is the best fit. In the first step, we assessed a single quadratic polynomial trajectory model. We conducted a quadratic two-trajectory model if the quadratic component in one trajectory model was significant. The model was repeated with a linear trajectory, if the quadratic term was not significant, to determine the BIC value. The BIC value of the appropriate two-trajectory model was compared to the BIC value of the appropriate one-trajectory model and the procedure was repeated by increasing the number of trajectories until the best-fitting model. We stopped the modeling once ΔBIC became a negative value. Given the possible confounding effect of varying donation intervals on ferritin levels, we also conducted the trajectory analyses with adjustment for time since the previous donation. To assess the effect of ferritin level distributions on the trajectory modeling the analyses were also conducted with log-transformed ferritin levels.

After assigning all donors to one of several trajectories, the characteristics of donors with differing ferritin trajectories were described as medians and interquartile ranges, or percentages in the case of proportions, stratified by gender. The characteristics of the donors in differing trajectories in each gender were compared using the Mann–Whitney U test for continuous variables and Pearson’s Chi-square test for categorical variables.

Associations between ferritin levels at a certain donation and the subsequent low iron level were assessed by generalized linear mixed models adjusted for age, sex, and time since the previous donation (a low iron was defined as ferritin levels <15 ng/mL, or low-Hb deferral with Hb levels <8.4 mmol/L for men or <7.8 mmol/L for women, corresponding with 135 and 125 g/L, respectively). Receiver operating characteristic (ROC) analysis was applied to compute the sensitivity and specificity for the prediction of subsequent low iron levels. The ROC curve is demonstrating the prediction of low iron levels at a subsequent donation attempt based on ferritin levels at the current donation (which could be any of the first 4 donations) in new whole blood donors; this curve was calculated based on specificity to detect true cases with low iron levels and 1 minus specificity for detecting cases with normal iron levels. The analyses were conducted in IBM SPSS Statistics for Windows, Version 26.0. Armonk, NY, USA: IBM Corp and Traj module in STATA version 16.1 for Windows (16.1, StataCorp LLC, College Station, TX, USA) 

## 3. Results

In total, 300 donors were included in the study, of whom 101 (33.7%) were male. Male donors, on average, made 4 donations within two years (median number of donations), and female donors 3, enabling the analysis of 1447 stored samples from successful donations. The median age of male donors was 29.96 years and of female donors was 24.62 years. Median levels of ferritin at baseline were 103.35 µg/L and 28.13 µg/L at the last donation for male donors. Among female donors, the median ferritin levels at first and last donations were 36.66 and 13.04 µg/L, respectively (Table 1).

The spaghetti plots in Figure 1 demonstrate the individual crude changes in ferritin levels by the number of donations. According to the smallest BIC and the logarithm of the Bayes factor, a two-group ferritin level trajectory model (separately) for male and female donors was very strongly favored. The findings were similar to log-transformed ferritin levels, and therefore results on non-transformed data are shown to enable easier interpretation. In male donors, the value of the logarithm of the Bayes factor was 18.46 (>10), therefore favoring the two-cluster model over a three-cluster model. In female donors, the value of the logarithm of the Bayes factor was 88.46 (>10), favoring the three-trajectory model over a four-cluster model; however, due to the small proportion of observations in the third cluster (<5%), the two-cluster model was more appropriate. The final analyses confirmed that models with two latent classes fit the data best, for both male and female donors, both with and without adjustment for donation intervals (Figure 2A,B). Class I, encompassing 42.9% of male donors and 87.7% of female donors, represents a linear reduction in ferritin levels, with a rather mild slope at earlier donations, later leading to steady ferritin levels. Despite a mild linear reduction in ferritin levels in class one, it is worth noticing that donors in this class of both genders have relatively low ferritin levels at baseline. In class two, baseline ferritin levels are relatively high, and there are more drastic reductions in ferritin levels over repeated donations. The donors in class one (with relatively low baseline levels of ferritin) are younger, have lower BMI, have lower ferritin levels at baseline and after two years, and have shorter donation intervals as compared to the donors in class two (Table 2).

The odds for iron deficiency, at any time point for the next donation, among donors with ferritin levels of <30 ng/mL, were 1.43 to 3.83-fold higher in comparison to those with ferritin levels of >30 ng/mL (Table 3). Based on the ROC curve of subsequent iron deficiency or low Hb detection at different ferritin levels, ferritin levels between 25 and 30 ng/mL showed both sensitivity and specificity to be above 70% (Figure 3).

## 4. Discussion

In this study among 300 new whole blood donors whose ferritin levels were measured retrospectively and repeatedly over donations for 2 years, we found two latent classes of ferritin trajectories for separate genders. All trajectories show declining ferritin levels, varying in baseline and slope. Additionally, we found that low ferritin levels (<30 ng/mL), if not acted upon, are significantly associated with a subsequent increased risk of returning to donate while iron-deficient (i.e., Ferritin levels >15 ng/mL), or having low Hb levels.

Generally, in male and female donors, it appears that all donors show some level of decline in ferritin levels, and the degree of this decline may depend on baseline ferritin levels. Based on previous studies showing a high prevalence of iron deficiency in repeat donors, and relatively long recovery of iron stores, this finding might not be very surprising [1,5,9,24]. While Hb levels may remain stable over consecutive donations, iron stores continue to be depleted, which remains unnoticed when ferritin levels are not measured [13,14]; however, previous findings were either based on single measurements rather than repeated measurements [13] or based on Hb levels rather than ferritin levels, which is a better indicator of iron status [25]. To our knowledge, the present study was the first study in which we followed donors with repeated ferritin measurements over time, thus enabling us to look into the shape of the ferritin trajectories over time, and whether or not these differ between individual donors. The finding that the decline in ferritin levels is steeper with higher baseline ferritin levels can probably be explained by hepcidin levels only being downregulated in case of an iron shortage, to increase iron absorption and release [26]. In other words, in donors with high iron availability, higher levels of hepcidin lead to blockage of ferroprotein, thereby inhibiting iron absorption in the gut and the release of stored iron into the circulation; this, in turn, reduces the pace of iron repletion in donors with high iron levels.

Among men, a minority of donors with high baseline ferritin levels showed relatively fast declines in ferritin levels. Among these donors, the levels of ferritin generally dropped to levels of iron deficiency after 6 to 10 donations. For the group with lower baseline ferritin levels, a less steep linear reduction was observed in ferritin levels, but also after 3–4 donations on average the level of iron deficiency was reached. Adjusting the models for donation intervals resulted in two trajectories with more similar starting points and slopes, indicating that the donation interval is often shorter in donors with high baseline ferritin levels and as a result show more steeply declining ferritin levels.

Female donors who were assigned to the trajectory with more steeply declining ferritin levels were older and had higher baseline ferritin levels than those in the other latent class. The higher baseline iron status among women of older age is in line with expectations and probably due to the cessation of menstrual blood loss after menopause [15]. Female donors with high baseline ferritin levels generally had ferritin levels higher than 15 ng/mL after 5 donations in two years. Nonetheless, those with lower baseline levels dropped to below this level after 4 donations. Adjusting for donation intervals resulted in a more gradual slope of reductions in ferritin levels.

In earlier studies, prediction models for Hb levels and low-Hb deferrals were developed [25,27]; these models included regression models, transition models, or a combination of these two approaches for predicting Hb levels based on routinely collected information on blood donors and their donations [25]. The findings from Donor InSight, a cohort study of donors at Sanquin aiming to increase insights into donor health and behavior, presented four different Hb trajectories among 5388 new donors [25]. The latter suggests that relying on a single trajectory for describing changes in Hb levels of donors through successive donations is not a suitable approach for dealing with the underlying changes in iron status among this population.

A major strength of the study is that ferritin levels were measured retrospectively at every donation over two years; this allowed for detecting trajectories of iron stores among donors, presenting both inter-and intra-individual differences in ferritin levels over repeated donations, and this enabled showing that low ferritin not only predicts subsequent low Hb but also iron deficiency through repeated measurements [13]. Yet, our study suffered from a limited sample size, due to which we were not able to provide any information on whether the identified subpopulations differed in their risk to develop low pre-donation hemoglobin levels.

The latter in turn, calls for future studies with larger sample sizes to verify these results. Another limitation, applying to any study on ferritin levels, is the fact that ferritin measurements, despite being calibrated in collaboration with other laboratories, are not standardized internationally; this may hamper external validity and thus the generalizability of ferritin levels and the cut-offs in particular.

Symptoms that are frequently related to whole blood donation-induced low iron include decreased physical endurance and work capacity, fatigue, and impairment in concentration, attention, and other cognitive functions, as well as restless leg syndrome and craving and consuming of non-nutritive substances (pica) [28,29,30,31,32,33,34,35,36]. Given the significant magnitude of undesired effects of blood donation-induced iron depletion, with, on the other hand, the high impact on donor availability when deferring too many donors, it is of high importance to determine appropriate cut-off levels for ferritin-guided donor deferrals. The results of our study showed ferritin levels of 25–30 ng/mL to have the highest sensitivity and specificity for subsequent low iron (defined as ferritin levels <15 ng/mL or Hb deferral).

Even though the ferritin trajectories seem to mainly depend on initial ferritin levels, the detected two different trajectories in both genders differ greatly in terms of age and BMI. In other words, individuals of a different age and BMI differ in terms of pre-donation ferritin levels as well as subsequent iron store repletion. Furthermore, higher initial ferritin levels were not accompanied by higher initial Hb levels, which provides further proof on Hb levels serving as an acute phase reaction marker among the donors with higher initial ferritin levels; moreover, based on previous findings, these groups might differ in terms of genetic predisposition to differences in iron metabolism and lifestyle behaviors [15,37,38,39]. In future studies, carefully modeling the interplay between iron stores and Hb levels, thereby also considering individual differences in demographics, lifestyle behaviors, and—if appropriate—environmental influences could help to better predict low iron and develop tailored donor iron management strategies.

Proposed strategies for donors at risk of low iron might be to increase donation intervals to prevent rapid reductions in iron stores through successive donations. Making 5 or 6 donations within 2 years might not result in iron deficiency or low-Hb deferrals in men and women with high baseline ferritin levels. As a proper strategy for maintaining iron stores among risk groups, it might be helpful to restrict donations to four per two years in this subgroup. Iron supplementation could aid in increasing donation intensities without inducing risks of low iron [40]. In addition, ferritin level monitoring could help to monitor and adjust individual donor strategies.

## 5. Conclusions

In conclusion, repeated ferritin measurements show depleting iron stores in all-new whole blood donors, the level of which depends on baseline ferritin levels. Tailored, less intensive donation strategies might help to prevent low iron in donors, and could be supported with ferritin monitoring and iron supplementation.

## Figures and Tables

**Figure 1 jcm-11-03581-f001:**
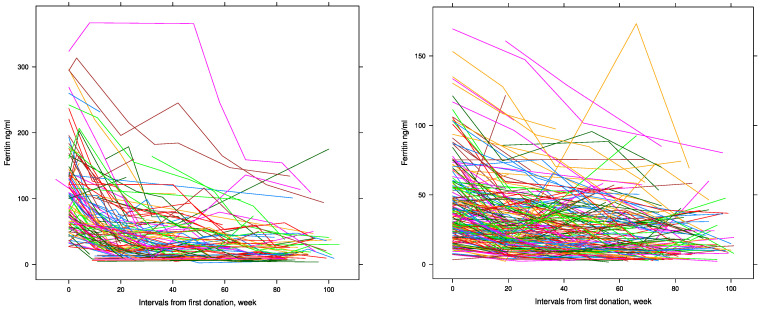
Spaghetti plots demonstrating ferritin levels over repeated donations per individual male (**left**) and female donor (**right**) in the FIND+ study by time since baseline, September 2017–September 2019.

**Figure 2 jcm-11-03581-f002:**
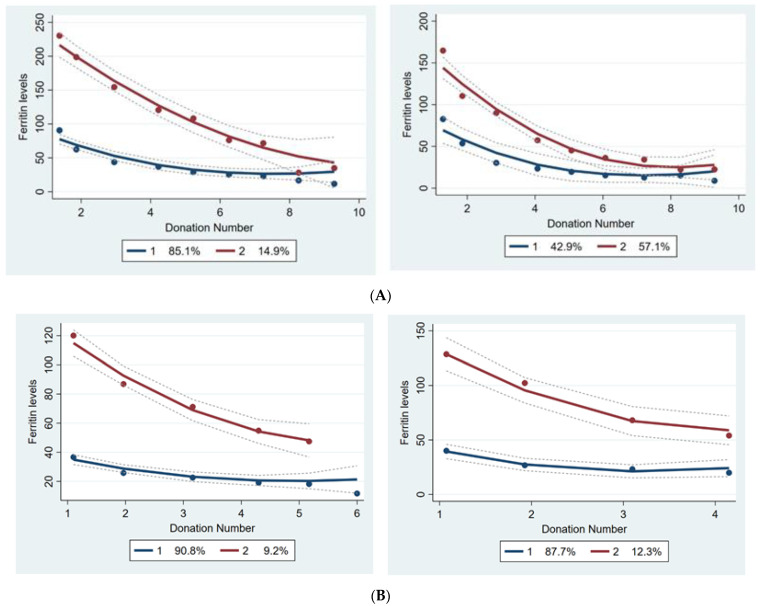
(**A**) Group-based trajectory analyses exhibit changes in ferritin levels by donation number in each latent class for male donors in unadjusted models (left) and models adjusted for time since the previous donation (right) in the FIND+ study September 2017–September 2019. Blue line (line No. 1) refers to class 1. Red line (line No. 2) refers to class 2. (**B**) Group-based trajectory analyses exhibiting changes in ferritin levels by donation number in each of the latent classes for female donors in unadjusted models (left) and models adjusted for time since the previous donation (right) in the FIND+ study, September 2017-September 2019. Dotted line represents the confidence interval for the levels of ferritin over donation times. Blue line (line No.1) refers to class 1. Red line (line No.2) refers to class 2.

**Figure 3 jcm-11-03581-f003:**
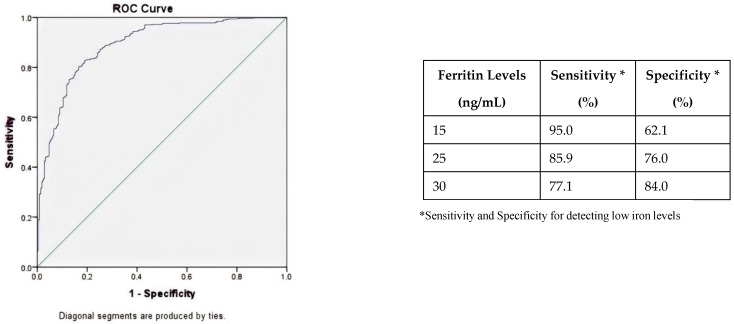
Receiver operating characteristic curve for predicting low iron (defined as ferritin <15 ng/mL or low-Hb deferral) at a subsequent donation attempt based on ferritin levels at any of the first 4 donations in new whole blood donors. Area under the curve is 0.89.

**Table 1 jcm-11-03581-t001:** Descriptive of donors included in the FIND+ dataset: new whole blood donors with more than two blood donations between September 2017 and September 2019 ^1^.

Variables	Male Donors (*n* = 101)	Female Donors (*n* = 199)
Age (years)	29.96 (25.08–38.12)	24.62 (22.58–29.60)
BMI (kg/m^2^)	24.22 (22.48–26.10)	23.14 (21.55–26.44)
Number of donations	4 (2–6)	3 (2–4)
Hb at screening visit (mmol/L)	9.30 (8.90–9.60)	8.30 (8.00–8.70)
Baseline ferritin levels (ng/mL) Ferritin at last donation ^2^ (ng/mL)	103.35 (56.77–137.76) 28.13 (17.80–43.24)	36.66 (23.21–56.59) 13.04 (10.63–29.30)
Inter-donation interval (weeks)	37.00 (19.00–63.00)	46.00 (24.00–69.00)
Donation in cold seasons (%)	47.1	48.2

^1^ The variables are presented as median and interquartile range or as stated otherwise. ^2^ Ferritin levels at 6th donation in men and 4th donation in women during the study period (September 2017–September 2019).

**Table 2 jcm-11-03581-t002:** Descriptive of donors in the FIND+ dataset by gender and group-based trajectory of the growth mixture model. ^1^

	Male Donors		Female Donors	
Variables	Class 1 (*n* = 87)	Class 2 (*n* = 14)	Class 1 (*n* = 181)	Class 2 (*n* = 18)
Age (years) *	28.11 (24.82–34.27)	39.43 (31.74–47.64)	24.42 (22.36–27.98)	35.00 (27.28–47.59)
BMI (kg/m^2^) *	23.86 (22.15–25.69)	27.46 (24.81–28.34)	22.85 (21.48–26.23)	25.18 (23.23–29.04)
Number of donations	4 (2–6)	4 (2–6)	3 (2–4)	3 (2–5)
Hb at screening visit (ng/mL)	9.40 (8.90–9.80)	9.05 (8.82–9.65)	8.30 (8.00–8.80)	8.35 (8.17–8.55)
Ferritin at screening visit (ng/mL) * Ferritin at last donation ^2^ (ng/mL) *	85.40 (56.10–120.24) 24.89 (14.16–31.10)	239.62 (181.63–294.61) 53.26 (43.78–133.74)	32.69 (20.34–48.90) 13.52 (6.30–28.46)	114.13 (102.55–135.31) 53.45 (39.01–68.89)

^1^ The variables are presented as median and interquartile range or as stated otherwise. ^2^ Ferritin levels at 6th donation in men and 4th donation in women during the study period (September 2017–September 2019). * Significantly different between class 1 and class 2 in male and female donors *(p*-values > 0.05) *p*-values are calculated using Mann–Whitney U test for continuous variables and Pearson’s Chi-square test for categorical variables.

**Table 3 jcm-11-03581-t003:** Associations between ferritin levels and subsequent low iron levels (ferritin levels <15 ng/mL or low-Hb deferral) at any of the first 4 donations over 895 whole blood donations, made by 300 donors in total.

Donor Ferritin Levels (ng/mL) *	N **	Odds Ratio	95% Confidence Interval	*p*-Value
Ferritin levels <15	99/193	3.83	3.19 to 4.48	<0.001
Ferritin levels 15–19.9	26/82	2.44	1.73 to 3.15	<0.001
Ferritin levels 20.1–24.9	15/62	2.10	1.31 to 2.89	<0.001
Ferritin levels 25–29.9	16/85	1.45	0.71 to 2.22	0.08
Ferritin levels >30	20/473	Ref	Ref	-

The odds ratios and confidence intervals are calculated using generalized linear mixed models adjusted for age, gender, and time since the previous donation. Low iron was defined as ferritin levels <15 ng/mL, or low-Hb deferral: Hb levels <8.4 mmol/L for men or <7.8 mmol/L for women, corresponding to 135 and 125 g/L, respectively. * Indicates the level of ferritin at subsequent donation. ** Number of donors with low iron levels at subsequent donation/total number of donations in the indicated ferritin category.

## Data Availability

Not applicable.

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
