# Peer review of "Ferritin Trajectories over Repeated Whole Blood Donations: Results from the FIND+ Study"

_jcm, 2022, doi:10.3390/jcm11133581_

Round 1

Reviewer 1 Report

With this work the authors continue their investigations of the iron and haemoglobin values in whole blood donors. The overall aim of such kind of investigations is to study the impact of blood loss associated with whole blood donation on the health status of the donors and their ability to further donate blood. The findings could help to develop better-tailored donor management strategies to prevent subsequent deferrals due to low pre-donation haemoglobin levels. The key finding of this study is the identification of subpopulations of donors with different ferritin trajectories over repeated donations. Although the results require confirmation by studies with larger numbers of blood donors, there is now some first evidence that beyond age and sex other factors may exist that predispose for different initial ferritin levels and iron load trajectories over repeated whole blood donations.

The second finding of this study that low ferritin levels are significantly associated with a subsequent increased risk of iron deficiency or low Hb levels in donors that regularly return to donate is not new. Unfortunately, the study provides no information (maybe due to the limited number of donors included) whether the identified subpopulations differed in their risk to develop low pre-donation haemoglobin levels. Please, work or comment on this.

In general, the authors should discuss in more detail the rationale and results of the present investigations in context of their previous research (e.g., Prinsze et al. Donation-induced iron depletion is significantly associated with low haemoglobin at subsequent donations. Transfusion 2021;61:3344–3352). Why didn’t the authors use the samples from the 5056 whole blood donors of the previous investigation for the present study?

Please, explain the study name FIND+.

This reviewer does not feel fully qualified to assess the applied statistical methods. Therefore, independent review of the results by a statistician is required. In particular, the receiver operating characteristic curve presented in Figure 3 needs more explanation. The relationship between the curves and the specificities and sensitivities shown in the little table is not fully clear.

Discussion, lines 40 to 42: Please, explain in more detail the hypothesis that the finding that the decline of ferritin levels is steeper with higher baseline ferritin levels is associated with hepcidin levels.

Discussion, line 62: Please, explain the term “Donor InSight”.

Table 3: What do the numbers given in the second column refer to? First number: low ferritin levels; and second number: low-Hb deferral?

Author Response

Comments from reviewer #1

We thank reviewer #1 for the critical review of our manuscript and constructive comments. Here we respond to each comment and show the changes we applied to our manuscript based on these remarks.

With this work the authors continue their investigations of the iron and haemoglobin values in whole blood donors. The overall aim of such kind of investigations is to study the impact of blood loss associated with whole blood donation on the health status of the donors and their ability to further donate blood. The findings could help to develop better-tailored donor management strategies to prevent subsequent deferrals due to low pre-donation haemoglobin levels. The key finding of this study is the identification of subpopulations of donors with different ferritin trajectories over repeated donations. Although the results require confirmation by studies with larger numbers of blood donors, there is now some first evidence that beyond age and sex other factors may exist that predispose for different initial ferritin levels and iron load trajectories over repeated whole blood donations.

Point 2. In general, the authors should discuss in more detail the rationale and results of the present investigations in context of their previous research (e.g., Prinsze et al. Donation-induced iron depletion is significantly associated with low haemoglobin at subsequent donations. Transfusion 2021;61:3344–3352). Why didn’t the authors use the samples from the 5056 whole blood donors of the previous investigation for the present study?

Reply. In DIS-III (the study by Prinsze et al), we cross-sectionally showed that higher total numbers of blood donations are significantly associated with lower ferritin levels. These measurements were, however, single measurements rather than repeated measurements. The present (FIND+) study was the first study in which we followed donors with repeated ferritin measurements over time, thus enabling us to look into the shape of the ferritin trajectories over time, and whether or not these differ between individual donors. Further explanations are now added to the manuscript, please see discussion:

Line 291-296. However, previous findings were either based on single measurements rather than repeated measurements [13] or based of Hb levels rather than ferritin levels which is a better indicator of iron status [25]. To our knowledge, the present study was the first study in which we followed donors with repeated ferritin measurements over time, thus enabling us to look into the shape of the ferritin trajectories over time, and whether or not these differ between individual donors.

Point 3. Please, explain the study name FIND+.

Reply. FIND+ stands for: Ferritin measurement IN Donors – donor Population LongitUdinal Study, this is now added to manuscript, line 79-80.

Point 4.This reviewer does not feel fully qualified to assess the applied statistical methods. Therefore, an independent review of the results by a statistician is required. In particular, the receiver operating characteristic curve presented in Figure 3 needs more explanation. The relationship between the curves and the specificities and sensitivities shown in the little table is not fully clear.

Reply. Thanks for your comment, the Receiver operating characteristic curve is demonstrating the prediction of low iron levels at a subsequent donation attempt based on ferritin levels at any of the first 4 donations in new whole blood donors. This curve is calculated based on specificity to detect true cases with low iron levels and 1- specificity for detecting cases with normal iron levels. The required clarifications is now added to the results section in the manuscript:

Lines 163-170. Receiver operating characteristic (ROC) analysis was applied to compute the sensitivity and specificity for the prediction of subsequent low iron levels. The ROC curve is demonstrating the prediction of low iron levels at a subsequent donation attempt based on ferritin levels at the current donation (which could be any of the first 4 donations) in new whole blood donors. This curve was calculated based on specificity to detect true cases with low iron levels and 1- specificity for detecting cases with normal iron levels.

Point 5. Discussion, lines 40 to 42: Please, explain in more detail the hypothesis that the finding that the decline of ferritin levels is steeper with higher baseline ferritin levels is associated with hepcidin levels.

Reply. In individuals with low iron availability, downregulated levels of hepcidin lead to increased iron absorption and release into the circulation, which in turn prevents rapid iron reduction in donors with low iron levels. Further explanation is now added to manuscript, please see

Lines 299-302. In other words, in donors with high iron availability, higher levels of hepcidin lead to blockage of ferroportin, thereby inhibiting iron absorption in the gut and the release of stored iron into the circulation. This in turn reduces the pace of iron repletion in donors with high iron levels.

Point 6. Discussion, line 62: Please, explain the term “Donor InSight”.

Reply. Donor InSight is the name of the cohort study aiming to increase insights into donor health and behavior, please see manuscript:

Lines 321-323. Donor InSight a cohort study of donors at Sanquin aiming to increase insights into donor health and behavior, …

Point 7. Table 3: What do the numbers given in the second column refer to? First number: low ferritin levels; and the second number: low-Hb deferral?

Reply. The required explanation is now added to the table 3:

Line 267. Number of donors with low iron levels at subsequent donation / total number of donors in the indicated ferritin category.

Reviewer 2 Report

This is a very nice paper, easy to follow and with good data on a very relevant topic. Although the statistics are more complex than I am capable to comprehend, I trust the authors and their expertise in this area.

I only have a few requests to the authors:

1.       Please state what is minimum required number of days between whole blood donations in the Netherlands.  Is it the same 56 days as it is in the United States? It is important to include that in the paper as a simple fact.  Probably in the Introduction.

2.       Page 2: Although I understand the following sentence (Hb level recovery in donors depends on the availability of iron [15, 16].), it seems to need something more such as “Previous studies have shown that …” Please check.

3.       I did not a reference range for ferritin in the paper.  Please add that where discussing levels of ferritin in donors.

4.       In general, I prefer to see data either in the text or in a table.  At the beginning of the Results section, some of the numerical data are in the text and table 1. Perhaps the text should summarize the findings and leave the data per se in the table only.

5.       Figures 2A and 2B are confusing because the lines are labeled 1 and 2 instead of Class 1 and Class 2 as used in the text.  If this is not what the graphs are representing, please clarify.

6.       Please explain how you choose the various ferritin level intervals shown in Table 3.  They seem so narrow considering the coefficient of variation of the test?

7.       Finally, towards the end of the discussion, the authors mention that both age and BMI influences on the two different ferritin trajectories are very important (Table 2), but they do not expand much on it except to discuss their findings in menopausal women. Maybe they thought that we all expect a 500 mL whole blood donation to have different effects in donors with small versus a large BMI and how they recover. I was left with the impression that as age increases so does the BMI and apparently ferritin levels as well; is it an age effect and/or obesity generating more inflammation? Looking at Table 2, higher ferritin levels in class 2 donors do not correlate with higher hemoglobin levels so it seems that it is probably acting as an acute phase reaction marker in this group. Could you expand on this issue?

Author Response

Comments from reviewer #3

We appreciate very much reviewer number 3 for commenting on our manuscript. Here we address the comment.

This is a very nice paper, easy to follow and with good data on a very relevant topic. Although the statistics are more complex than I am capable to comprehend, I trust the authors and their expertise in this area.

I only have a few requests for the authors:

Point 1. Please state what is the minimum required number of days between whole blood donations in the Netherlands. Is it the same 56 days as it is in the United States? It is essential to include that in the paper as a simple fact. Probably in the Introduction.

Reply. In the Netherlands, the general minimum interval between whole blood donations for women is 122 days and in men is 56 days, the required information is now added to the introduction please see lines 62-64.

Point 2. Page 2: Although I understand the following sentence (Hb level recovery in donors depends on the availability of iron [15, 16].), it seems to need something more such as “Previous studies have shown that …” Please check.

Reply. The required edition is applied, please see manuscript lines 60-61.

Point 3. I did not a reference range for ferritin in the paper.Please add that where discussing levels of ferritin in donors.

Reply. Ferritin levels > 15 ng/ml are regarded as iron deficiency, it is now indicated in the manuscript, line 289

Point 4. In general, I prefer to see data either in the text or in a table. At the beginning of the Results section, some of the numerical data are in the text and table 1. Perhaps the text should summarize the findings and leave the data per se in the table only.

Reply. The required editions are done and now result in section lines 176-185 only summarize the findings.

Point 5. Figures 2A and 2B are confusing because the lines are labeled 1 and 2 instead of Class 1 and Class 2 as used in the text. If this is not what the graphs are representing, please clarify.

Reply. lines are labeled 1 and 2 are the same as Class 1 and Class 2 as used in the text, a legend is now added to Figures 2A and 2B to clarify this.

Point 6. Please explain how you choose the various ferritin level intervals shown in Table 3. They seem so narrow considering the coefficient of variation of the test?

Reply. The ferritin levels are chosen based on cutoffs of WHO (>15 ng/ml, see Reference number.1) and Sanquin (ferritin levels >30 ng/ml is Sanquin Blood Bank policy). Intervals between 15 to 30 ng/ml were chosen with a 5 ng/ml increase in ferritin levels at each intervals. The odds for iron deficiency, among donors with ferritin levels of <25ng/ml, had significantly higher odds of subsequent iron deficiency, which means the latter range could be a safer cut-off for whole blood donors to reduce the risk of post donation iron deficiency.

Point 7. Finally, towards the end of the discussion, the authors mention that both age and BMI influences on the two different ferritin trajectories are very important (Table 2), but they do not expand much on it except to discuss their findings in menopausal women. Maybe they thought that we all expect a 500 mL whole blood donation to have different effects on donors with small versus a large BMI and how they recover. I was left with the impression that as age increases so do the BMI and ferritin levels as well; is it an age effect and/or obesity generating more inflammation? Looking at Table 2, higher ferritin levels in class 2 donors do not correlate with higher hemoglobin levels so it seems that it is probably acting as an acute phase reaction marker in this group. Could you expand on this issue?

Reply. Thanks for your comment, based on our findings, individuals with different of age and BMI differ in terms of initial ferritin levels as well as iron store replation. Furthermore, higher initial ferritin levels was not accompanied by higher initial Hb levels, which provides further proof on Hb levels serving as an acute phase reaction marker among the terajecory with higher intitial ferritin levels.

The required explanation is now added to discussion, please see lines 362-365.

Reference:

  1. Sweegers, M.G., et al., Ferritin measurement IN Donors-Effectiveness of iron Monitoring to diminish iron deficiency and low haemoglobin in whole blood donors (FIND'EM): study protocol for a stepped wedge cluster randomised trial. Trials, 2020. 21(1): p. 823.

Reviewer 3 Report

The study explores the ferritin trajectories observed in 300 donors over an average number of 4 donations. On the basis of different trajectories, two groups of donors are identified in both genders. and the odds for iron deficiency (ferritin < 15) or donation deferral for low Hb at the next donation is calculated. 

The conclusions have great relevance for donor health and can be helpful in the daily management of donations in blood banks. Indeed I would congratulate the authors on their study.   

I have only one suggestion. Did the authors try to incorporate different ferritin levels together with additional variables with a conceivable effect on iron status (age, gender, BMI, donation interval, or the number of donations) in a multivariate model to predict the risk for deferral at the next donation?  It could be possible to design an algorithm with higher specificity and sensitivity as compared to ferritin only? It could be a useful tool in the daily management of donors. 

Author Response

Comments from reviewer #2

We appreciate very much reviewer number 2 for commenting on our manuscript. Here we address the comment.

The study explores the ferritin trajectories observed in 300 donors over an average number of 4 donations. On the basis of different trajectories, two groups of donors are identified in both genders. and the odds for iron deficiency (ferritin < 15) or donation deferral for low Hb at the next donation is calculated. 

The conclusions have great relevance for donor health and can be helpful in the daily management of donations in blood banks. Indeed I would congratulate the authors on their study. 

Point 1. I have only one suggestion. Did the authors try to incorporate different ferritin levels together with additional variables with a conceivable effect on iron status (age, gender, BMI, donation interval, or the number of donations) in a multivariate model to predict the risk for deferral at the next donation? It could be possible to design an algorithm with higher specificity and sensitivity as compared to ferritin only? It could be a useful tool in the daily management of donors. 

Reply. Thank you for congratulating us on our study. Regarding prediction modeling: that is indeed something that we are working on as well, but this work is still in progress. In addition, as you may know, prediction modeling is serving a different type of aim and requires a different type of methodology and is therefore outside the scope of the current manuscript.